# The Role of Pyrazolopyridine Derivatives on Different Steps of Herpes Simplex Virus Type-1 *In Vitro* Replicative Cycle

**DOI:** 10.3390/ijms23158135

**Published:** 2022-07-23

**Authors:** Milene D. Miranda, Otávio Augusto Chaves, Alice S. Rosa, Alexandre R. Azevedo, Luiz Carlos da Silva Pinheiro, Vinicius C. Soares, Suelen S. G. Dias, Juliana L. Abrantes, Alice Maria R. Bernardino, Izabel C. P. Paixão, Thiago Moreno L. Souza, Carlos Frederico L. Fontes

**Affiliations:** 1Laboratório de Morfologia e Morfogênese Viral, Instituto Oswaldo Cruz, Fundação Oswaldo Cruz, Rio de Janeiro 21040-360, RJ, Brazil; srosa.alice@gmail.com; 2Laboratório de Estrutura e Regulação de Proteínas, Instituto de Bioquímica Médica, Programa de Biologia Estrutural, Universidade Federal do Rio de Janeiro, Rio de Janeiro 21941-902, RJ, Brazil; lourenco.abrantes@gmail.com; 3Laboratório de Imunofarmacologia, Instituto Oswaldo Cruz (IOC), Fundação Oswaldo Cruz (Fiocruz), Rio de Janeiro 21040-360, RJ, Brazil; otavioaugustochaves@gmail.com (O.A.C.); cardosodante42@gmail.com (V.C.S.); suelen.sgdias@gmail.com (S.S.G.D.); thiago.moreno@fiocruz.br (T.M.L.S.); 4Instituto Nacional para Ciência e Tecnologia sobre Inovação em Doenças de Populações Negligenciadas (INCT/IDPN), Centro de Desenvolvimento Tecnológico em Saúde (CDTS), Fundação Oswaldo Cruz (Fiocruz), Rio de Janeiro 21040-360, RJ, Brazil; 5Centro Universitário UniLaSalle-RJ, Departamento de Engenharia de Produção, Niterói 24240-030, RJ, Brazil; prof.alexandre.azevedo@soulasalle.com.br; 6Departamento de Ciências, Faculdade de Formação de Professores, Universidade do Estado do Rio de Janeiro, São Gonçalo 24435-005, RJ, Brazil; luiz.pinheiro@uerj.br; 7Programa de Imunologia e Inflamação, Universidade Federal do Rio de Janeiro, Rio de Janeiro 21941-902, RJ, Brazil; 8Instituto de Ciências Biomédicas (ICB), Universidade Federal do Rio de Janeiro, Rio de Janeiro 21941-902, RJ, Brazil; 9Laboratório de Virologia Molecular, Instituto de Biologia, Programa de Pós-graduação em Ciências e Biotecnologia, Programa em Neurologia e Neurociências e Programa de Biotecnologia Marinha, Universidade Federal Fluminense, Niterói 24210-201, RJ, Brazil; izabeluff@gmail.com; 10Instituto de Química, Departamento de Química Orgânica, Programa de Pós-graduação em Química, Universidade Federal Fluminense, Niterói 24020-007, RJ, Brazil; bernardinoamr@gmail.com

**Keywords:** pyrazolopyridine, antiviral, molecular docking, HSV-1, ICP-27, gD

## Abstract

Herpes simplex virus type-1 (HSV-1) infection causes several disorders, and acyclovir is used as a reference compound. However, resistant strains are commonly observed. Herein, we investigate the effects of *N*-heterocyclic compounds (pyrazolopyridine derivatives), named **ARA-04**, **ARA-05**, and **AM-57**, on HSV-1 in vitro replication. We show that the 50% effective concentration (EC_50_) values of the compounds **ARA-04**, **ARA-05**, and **AM-57** were 1.00 ± 0.10, 1.00 ± 0.05, and 0.70 ± 0.10 µM, respectively. These compounds presented high 50% cytotoxic concentration (CC_50_) values, which resulted in a selective index (SI) of 1000, 1000, and 857.1 for **ARA-04**, **ARA-05**, and **AM-57**, respectively. To gain insight into which step of the HSV-1 replication cycle these molecules would impair, we performed adsorption and penetration inhibition assays and time-of-addition experiments. Our results indicated that **ARA-04** and **ARA-05** affected viral adsorption, while **AM-57** interfered with the virus replication during its α- and γ-phases and decreased ICP27 content during initial and late events of HSV-1 replication. In addition, we also observed that **AM-57** caused a strong decrease in viral gD content, which was reinforced by in silico calculations that suggested **AM-57** interacts preferentially with the viral complex between a general transcription factor and virion protein (TFIIBc-VP16). In contrast, **ARA-04** and **ARA-05** interact preferentially in the proteins responsible for the viral adsorption process (nectin-1 and glycoprotein). Thus, our results suggest that the 1*H*-pyrazolo[3,4-*b*]pyridine derivatives inhibit the HSV-1 replicative cycle with a novel mechanism of action, and its scaffold can be used as a template for the synthesis of promising new molecules with antiviral effects, including to reinforce the presented data herein for a limited number of molecules.

## 1. Introduction

Herpes simplex virus type 1 (HSV-1) is an *alphaherpesvirus*, whose virions consist of a large, double-stranded DNA genome packaged within an icosahedral capsid encased by a proteinaceous tegument and a lipidic envelope composed of various glycoproteins [1,2]. Most people have their first contact with HSV-1 during childhood, and if HSV-1 succeeds in invading the orolabial mucosa, they acquire a lifelong persistent infection, in which the viruses can establish latency in nervous system cells and even reactivate depending on the immune status of the human host. Recent estimates indicate 63–90% seropositivity among adults worldwide [3,4].

The HSV-1 replicative cycle has been extensively described [5,6]. HSV-1 entry into host cells is an intriguing process that requires a fusion apparatus protein core such as glycoprotein D (gD), gB, and heterodimer gH/gL [7,8,9,10]. VP16, a viral tegument protein, transactivates the expression of immediate-early (IE; α) genes. Once inside the cell, the HSV-1 replicative cycle occurs in a concerted cascade divided into three phases [9].

First, IE gene expression occurs [11]. HSV-1 immediate-early proteins, such as infected-cell protein 0 (ICP0), ICP27, and ICP4, regulate the subsequent steps of viral replication and affect normal host cell function [12,13]. ICP0 is known to be an E3-ubiquitin ligase that counteracts host repression of the viral genome, DNA damage responses, and host-driven chromatin repression of the viral genome [14]. ICP27 modulates 3′ processing, splicing, and exportation of mRNA [12,15,16]. ICP4 is the main viral transcriptional factor, stimulating the transcription of early and late genes, while ICP22 promotes transcription elongation of viral genes [17]. Second, delayed-early (DE; β) gene transcripts, such as viral helicase (UL-8), DNA polymerase (UL-30), and thymidine kinase, are expressed, and viral DNA is synthesized. Finally, after viral DNA replication, late (L; γ) genes, which encode HSV-1 structural proteins, such as gD, gB, gC, and ICP5, are expressed [5,6]. Among the most important HSV-1 proteins, the immediate-early protein ICP27 plays a key role during viral replication. It has been shown that this viral protein affects both viral and host cell gene expression, being crucial for the efficient expression of DE and L genes [16,18,19]. Another HSV-1 protein vital for viral replication is glycoprotein D (gD) [5,9]. This glycoprotein has structural roles and serves as a binding point to anchor interactions with three alternative receptors for virus entry: nectin-1, herpesvirus entry mediator (HVEM), and modified heparan sulfate [20,21]. Viral gD is also one of the most conserved proteins among the *Herpesviridae* family and constitutes the core apparatus of the viral fusion protein [7,8,22].

The treatment of HSV-1-infected individuals and various antiherpetic compounds have been widely investigated, including acyclovir (ACV), ganciclovir, and penciclovir, and also their prodrugs valacyclovir, valganciclovir, and famciclovir [23]. Nonetheless, the selective emergence of drug-resistant HSV-1 strains may result from prolonged administration of these agents to immunocompromised patients. Therefore, drug repurposing and the search for new antiviral compounds, particularly with novel mechanisms of action, is a critical aim [24,25].

Among the various compounds with biological activities, the 1*H*-pyrazolo[3,4-*b*]pyridine system is known to have anticancer, antimalarial, anti-inflammatory, analgesic, antibacterial, and antiprotozoal activities [26,27,28,29,30,31,32]. Moreover, some chemical modifications confer to the biological activity of these derivatives against Alzheimer’s disease and the HIV-1 enzyme reverse transcriptase (RT) [33,34]. In fact, the antiviral activity of the 1*H*-pyrazolo[3,4-*b*]pyridine and thieno[2,3-*b*]pyridine derivative systems was described by our group [35,36,37,38,39].

Our group demonstrated the inhibition of HSV-1 cytopathic effects by 1*H*-pyrazolo[3,4-*b*]pyridine derivatives [38]. However, the mechanism of action towards this virus replication remained to be characterized. Thus, to understand the main active derivatives against the mode of action of HSV-1 replication (Figure 1), we performed a set of in vitro studies to determine the 50% effective concentration (EC_50_) and 50% cytotoxic concentration (CC_50_) of these compounds, as well as their effects on HSV-1 adsorption, penetration, and virus replicative cycle. To offer a reasonable, molecular-level explanation of the mechanism of the assayed compounds, in silico calculations via molecular docking were carried out with the main targets identified in the experimental assays. We found that **ARA-04**, **ARA-05**, and **AM-57** were strong inhibitors of HSV-1 replication and fully capable of diminishing the content of some important viral proteins.

## 2. Results and Discussion

### 2.1. Inhibition Profile of HSV-1 by Pyrazolopyridine Derivatives

We have already described the synthesis and antiviral activity of some 1*H*-pyrazolo[3,4-*b*]pyridine derivatives [38]. Here, we named the studied compounds **ARA-04**, **ARA-05**, and **AM-57** (Figure 1). First, we determined the potency of the studied compounds. As shown in Table 1, the 50% effective concentration (EC_50_) values for **ARA-04** and **ARA-05** were about 1 µM, while for **AM-57**, it was 0.70 ± 0.10 µM. These values were comparable to the EC_50_ value for acyclovir (ACV, positive control). Importantly, the CC_50_ values for **ARA-04** and **ARA-05** were 1000 µM, while for **AM-57**, it was 600 µM. We calculated the selective index (SI) values based on the ratio of CC_50_ and EC_50_, which were equal to 1000 for **ARA-04** and **ARA-05** and 857.01 for **AM-57**, again comparable to the SI value for ACV [40].

Considering that pyrazolo[3,4-*b*]pyridine derivatives inhibit HSV-1 replication, and it has been shown that these compounds are non-nucleoside analog inhibitors of the HIV-1 enzyme reverse transcriptase [34,41], the structure of these derivatives could be an interesting model to develop novel drugs that could simultaneously target HIV-1 and HSV-1 replication since HSV-1 is known to be an opportunistic pathogen often found in HIV-infected patients [42].

### 2.2. Effects of Pyrazolopyridine Derivatives on Infected Cells

To gain insight into pyrazolopyridine’s inhibitory effect on HSV-1 replication, we analyzed the impact of antiviral treatment during different moments of the virus life cycle. To monitor if the antiviral activity was already present during adsorption, cells were infected and treated simultaneously for 1 h in an ice-cold bath at 4 °C because, at this condition, viral particles can bind but are unable to penetrate the host cells. After this period, cells were washed with ice-cold PBS to remove unbounded viruses, and the temperature was raised to 37 °C. After a temperature shift, the previously bound viral particles may enter the cells and produce plaques after 72 h p.i.

The compounds **ARA-04** and **ARA-05** significantly reduced the number of plaques when virus infection and treatment were synchronized at 4 °C (Figure 2A). **AM-57** was inactive under these conditions (Figure 2A). For comparison, we used heparin as a positive control at 1 µg/mL because it competes with heparan sulfate on the cell surface for HSV-1 adsorption (Figure 2A) [43,44].

In another set of experiments, cells were infected at 4 °C but without treatment. After 1 h, cells were washed with ice-cold PBS to remove unbounded viruses, and the temperature was shifted to 37 °C to allow virus penetration for an additional 1 h period. During this step, associated with virus penetration, treatments with **ARA-04**, **ARA-05**, or **AM-57** were performed. After 72 h, PFU were scored. No effect could be attributed to the inhibition of HSV-1 penetration (Figure 2B).

To gain insight into whether the HSV-1 replicative cycle would be impaired by the **ARA-04**, **ARA-05**, and **AM-57** compounds, a time-of-addition assay was performed because the synthesis of most HSV-1 α-, β-, and γ- proteins has been with particular time frames in Vero cells [9]. Vero cells were infected with HSV-1 for 1 h at 37 °C and treated with the compounds at a concentration equivalent to their EC_50_ concentrations since, at this level, 50% replication inhibition can be obtained without observing possible nonspecific effects. We found that the antiherpetic activity of **AM-57** was preserved when the drug was added to HSV-1-infected cells in the time frames from 0–3 h (Figure 3A) to 6–20 h after infection (Figure 3C). These results, along with the absence of anti-HSV-1 activity by AM-57 in Figure 2, could suggest that such a molecule may act both at the initial and late steps of viral replication, which overlaps with the peak of α- and γ-protein syntheses. The compounds **ARA-05** and **ARA-04** showed marginal effects when added post-infection/adsorption (Figure 3). For comparison, ACV, which inhibits HSV-1 DNA synthesis, a β-phase event, mainly affected the viral life cycle in the time frame from 3 to 6 h (Figure 3B).

Among the tested compounds, **AM-57** seems to have a bimodal effect on the HSV-1 replicative cycle, an activity that is desired among novel compounds in the literature [45,46,47]. Moreover, since there is a gap of time between the initial and late effects of this compound on the HSV-1 replication cycle, these two activities could be independent of each other.

Since the pyrazolopyridine compound blocks HSV-1 replication, considering it has been shown that this virus replication causes an interruption of host cell protein synthesis [2,6,47,48], and also that **AM-57** inhibition on the HSV-1 life cycle is temporally associated with the α- and γ-steps of viral replication, we monitored the levels of ICP27 and gD upon AM-57 treatment as proxies of these phases [47,48]. Vero cells were infected for 1 h at 37 °C and then treated with the compounds at their EC_50_ values during the time frames of 0–3 or 6–20 h p.i. After that, the cells were harvested, and the samples were submitted to SDS/PAGE electrophoresis and immunoblotting assays. When HSV-1-infected Vero cells were treated with **AM-57** during the peak of proteins from the α-phase (0–3 h p.i.) of HSV-1 replication, an 89.6% decrease in ICP27 content was observed (Figure 4A,B). In addition, when cells were treated with **AM-57** from 6 to 20 h p.i., a complete suppression of ICP27 content was observed (Figure 4C,D). These results indicate that **AM-57** may modulate the immediate-early and late events of HSV-1 replication [12,15,18,49]. Interestingly, **AM-57** lacks activity from 3 to 6 h p.i. (Figure 3B) because ICP27 synthesis/content could be turned off during this period, or such a protein would be in association with other cellular/viral molecules that would probably make ICP27 inaccessible to this compound [12,49].

The levels of gD from the HSV-1-infected cells treated with **AM-57** were completely reduced, reinforcing the notion that the late phase of the HSV-1 life cycle could be affected by this treatment (Figure 4E,F). **AM-57** does not necessarily act on ICP27 or gD directly. It is possible that AM-57 more broadly could affect the α- and γ-steps of viral replication.

### 2.3. In Silico Calculations for the Main Targets of Pyrazolopyridine Derivatives

We conducted in silico calculations, via molecular docking, to obtain insights into the molecular level of the **ARA-04**, **ARA-05**, and **AM-57** antiviral mechanism by evaluating the interaction mode and binding capacity between each pyrazolopyridine derivative with the proteins responsible for the viral adsorption process (glycoprotein and nectin-1 (gD-nectin-1)) [10] and one of the main proteins responsible for the ICP27 content (isolated proteins and complex composed by a general transcription factor and virion protein, TFIIBc-VP16) [50,51]. Table 2 summarizes the docking score values for each case. For GOLD 2020.2 software, more positive score values indicate better interactions [52]. Thus, in silico results suggested that **ARA-04** and **ARA-05** interact preferentially with the proteins responsible for the viral adsorption process, mainly gD, e.g., 55.8 and 46.8 for **ARA-04** into gD and nectin-1, respectively. On the other hand, **AM-57** interacts preferentially with the proteins responsible for the ICP27 content, mainly in the complex TFIIBc-VP16 (in this case, the docking score value for **AM-57** is about 5 times higher than that for **ARA-04** and **ARA-05**). These in silico results corroborate the experimental data that described different mechanisms between **ARA** compounds (inhibit the viral adsorption process) and **AM-57** (decrease ICP27 content). A combination of biochemical and biophysical assays, e.g., thermal shift surface plasmon resonance and, most importantly, structural experiments, e.g., X-ray-diffraction and nuclear magnetic resonance, should be performed to further clarify how **ARA-04** and **ARA-05** could target gD or nectin-1, as well as how **AM-57** could target TFIIBc-VP16, TFIIBc, or VP16.

Figure 5 and Figure 6 depict the binding pose of **ARA-04**, **ARA-05**, and **AM-57** into the gD-nectin-1 and TFIIBc-VP16 structure, respectively. According to in silico calculations, **ARA-04** and **ARA-05** have a similar binding pose for gD-nectin-1, while **AM-57** is less buried in the protein pocket, mainly in the case of gD (Figure 5B,C), directly impacting the number of amino acid residues that stabilize the interactive profile of each compound. In this case, interactions via van der Waals forces were detected.

In line with the inhibition of the α- and γ-steps of viral replication, by means of in silico assays, **AM-57** impairs the system TFIIBc-VP16. **AM-57** strongly interacts with the key amino acid residues of the TFIIBc/VP16 interface (Figure 6B–D), e.g., for **AM-57**, *t*-stacking interactions with Tyr-165 (1.9 Å), van der Waals forces with Phe-195 (3.4 Å), and hydrogen bonding with Lys-196, Arg-286, and Arg-290 (2.9, 2.3, and 3.7 Å, respectively) were detected, while for **ARA-04** and **ARA-05**, interactions via van der Waals forces were detected (Tyr-165, Phe-195, and Arg-286 within a distance of 2.7, 2.1, and 3.4 Å, respectively), and only one hydrogen bonding interaction was detected with Tyr-293 (1.9 Å). Therefore, the mechanistic hypothesis for **AM-57** is that this pyrazolopyridine compound might competitively interact with TFIIBc, inhibiting its interaction with the substrate VP16 to not form the complex TFIIBc-VP16. As a consequence, the subsequent steps of the virus life cycle would also be impaired.

## 3. Materials and Methods

### 3.1. Synthesis and Stock Solutions of 1H-Pyrazolo[3,4-b]pyridine Derivatives

The compounds ethyl 4-((5-methylpyridin-2-yl)amino)-1-phenyl-1*H*-pyrazolo[3,4-*b*]pyridine-5-carboxylate (**ARA-04**), ethyl 4-((6-methylpyridin-2-yl)amino)-1-phenyl-1*H*-pyrazolo[3,4-*b*]pyridine-5-carboxylate (**ARA-05**), and 6-chloro-9-fluoro-3-phenyl-3*H*-benzo[*b*]pyrazolo[3,4-*h*][1,6]naphthyridine (**AM-57**) (Figure 1) are synthetic derivatives of the 1*H*-pyrazolo[3,4-*b*]pyridine system. The synthesis of these compounds was previously described by us [38]. The derivatives were diluted in dimethylsulfoxide (DMSO) 100%, aliquoted, and stored at –20 °C. A maximum of 3 freezing and thawing cycles were performed to avoid compound degradation [54]. The DMSO concentrations during the assays were below 0.1%, in which case it can be considered not significantly cytotoxic [55,56].

### 3.2. Cells and Virus

Vero cells (African green monkey kidney cells, ATCC) were grown in Dulbecco’s modified Eagle’s medium (DMEM; GIBCO, Life Technologies, Grand Island, NY, USA) supplemented with 5% fetal bovine serum (FBS; HyClone, Logan, UT, USA), 100 U/mL penicillin, 100 µg/mL streptomycin, and incubated at 37 °C in 5% CO_2_. For virus stock preparation, Vero cells were infected with HSV-1 (KOS strain) [57] at a multiplicity of infection (MOI) of 0.1. Twenty-four hours post-infection (h.p.i.), the cells were lysed after three rounds of freezing and thawing, centrifuged at 1500× *g* at 4 °C for 20 min, and the resulting supernatant was pooled and stored at –70 °C for further studies [58].

### 3.3. Cytotoxic Assays

Vero cell monolayers (10^4^ cells/well) in 96-well plates were treated for 72 h with increasing amounts (312.5–2500 µM) of the tested compounds. Then, 5 mg/mL MTT (3-(4,5-dimethylthiazol-2-yl)-2,5-diphenyltetrazolium bromide, Sigma) diluted in 1 × PBS was added to the cells, following the manufacturer’s protocol. After 4 h at 37 °C, 10% SDS was added. Following incubation for 2 h at 37 °C, the plates were read in a spectrophotometer at 570 nm. All the compounds were resuspended in DMSO 100% for in vitro testing. In the assays, the final DMSO concentrations were equal to or lower than 1% (*v*/*v*), diluted in DMEM, and not toxic to the cells [59]. The 50% cytotoxic concentration (CC_50_) was calculated through a nonlinear regression analysis of the dose–response curves created from the data with GraphPad Prism 8 software.

### 3.4. Yield-Reduction Assays and Virus Titration

Vero cell monolayers in 24-well plates (4 × 10^4^ cells/well) were infected with HSV-1 at an MOI of 1 for 1 h at 37 °C. Then, the cells were washed, and the compounds were added at semilog concentrations (0.1–10 µM) in DMEM with 5% FBS. In every assay, we chose to use acyclovir (ACV; Sigma-Aldrich/Merck, St. Louis, MO, USA, >99% pure) as a positive control. After 24 h, the cells were lysed, centrifuged for clarification, and virus titers in the supernatant were determined by plaque-forming assays in Vero cell monolayers using 6-well plates (3 × 10^5^ cells/well). In this assay, Vero cells were exposed to supernatants from yield-reduction assays for 1 h at 37 °C. Next, residual viruses were discarded, and carboxymethylcellulose solution (DMEM-HG 1×, 1.0% carboxymethylcellulose, and 2% fetal bovine serum) was added. After 72 h at 37 °C, the cytopathic effect (CPE) was analyzed under an optical microscope, and 10% formalin was added to fix the cells. After 3 h, the solution was harvested, plaques were colored with 0.4% bromophenol blue, and the virus titers were calculated by scoring the plaque-forming units (PFU) [38,39,60]. The 50% effective concentration (EC_50_) was obtained through a nonlinear regression analysis of the dose–response curves generated from the data using GraphPrism 8 software. The selectivity indexes (SIs) for each assayed compound were calculated through the ratio between the CC_50_ and EC_50_ values.

### 3.5. Adsorption and Penetration Inhibition Assays

To test whether **ARA-04**, **ARA-05**, or **AM-57** could inhibit virus adsorption, Vero cells (3 × 10^5^ cells/well) grown in 6-well plates were infected with 200 PFU/well of HSV-1 and treated with each compound EC_50_ at 4 °C for 1 h. Then, cells were washed and covered with overlay medium, and the temperature was increased to 37 °C for 72 h [61]. After that, plaques were analyzed, as described in the plaque-forming assay. For this assay, heparin at 1 µg/mL (Sigma-Aldrich/Merck, St. Louis, MO, USA, >99% pure) was used as a positive control.

HSV-1 penetration assays were also performed, in which the same protocol mentioned above was used with slight modifications. In brief, Vero cells (3 × 10^5^ cells/well) were infected with 200 PFU/well of HSV-1 at 4 °C for 1 h. Next, the temperature was raised to 37 °C in the presence of **ARA-04**, **ARA-05**, or **AM-57**; 1 h after this step, cells were washed with PBS supplemented with glycine at pH 2.2. After this step, cells were covered with overlay medium, and at 72 h p.i., the PFUs were counted as described [62].

### 3.6. Time-Of-Addition Assays

To evaluate whether the addition of the compounds could be delayed without losing their capacity to inhibit HSV-1 replication and thus identify which period of the HSV-1 replication cycle is affected by these compounds, time-of-addition assays were performed. For this purpose, Vero cells in 6-well plates were infected with 200 PFU/well for 1 h at 37 °C. Then, monolayers were washed, and the derivatives were added at their respective EC_50_ at different hours p.i. [61]. After that, the medium containing the compounds was removed, and cells were covered with overlay medium. At 72 h p.i., the plaques were counted. As a control, the reference compound (ACV) at 1.1 µM (EC_50_) was analyzed in parallel.

### 3.7. Immunoblotting

To investigate whether **AM-57** can change ICP27 or gD expression, immunoblotting experiments were performed, as described [63,64]. In brief, Vero cells in 25 cm^2^ flasks (1 × 10^6^/flask) were infected with HSV-1 at an MOI of 1 for 1 h at 37 °C. Then, cells were washed and treated with the derivative at 0.70 µM at different times after virus infection. Next, cellular proteins were extracted using ice-cold lysis buffer (pH 8.0) (1% Triton X-100, 2% SDS, 150 mM NaCl, 10 mM HEPES, 2 mM EDTA plus protease inhibitor cocktail, Roche). Vero lysates were boiled at 100 °C for 5 min in the presence of Laemmli sample buffer (pH 6.8) (20% β-mercaptoethanol, 370 mM Tris base, 160 μM bromophenol blue, 6% glycerol, 16% SDS). Then, 20 μg of protein was submitted to electrophoresis on SDS-containing 8% polyacrylamide gel (SDS-PAGE). After that, the separated proteins were transferred to nitrocellulose membranes and incubated in blocking buffer (5% nonfat milk, 50 mM Tris-HCl, 150 mM NaCl, and 0.1% Tween 20). In the next step, the membranes were probed overnight with the antibodies anti-ICP27 (Santa Cruz Biotechnology, Dallas, TX, USA, sc-69806), anti-gD (Santa Cruz Biotechnology, sc-21719), or anti-β-actin (Sigma, Louis, MO, USA, A1978). After the washing steps, the HRP-conjugated secondary antibodies were used to interact with primary antibodies. The blocking buffer was used to dilute all antibodies. Additionally, the detections were performed using SuperSignal Chemiluminescence (GE Healthcare, Chicago, IL, USA). The densitometries were analyzed using Image Studio Lite Ver 5.2 software (LI-COR Biosciences, Lincoln, NE, USA).

### 3.8. In Silico Procedure

The 3D chemical structure of **ARA-04**, **ARA-05**, and **AM-57** was built and energy-minimized by density functional theory (DFT) via Spartan’18 software (Wavefunction Inc., Irvine, CA, USA). The crystallographic structure for gD-nectin-1, TFIIBc-VP16, and VP16 was obtained from the Protein Data Bank (PDB): 3SKU, 2PHG, and 16VP, respectively [50,51,53]. GOLD 2020.2 software (Cambridge Crystallographic Data Center Software Ltd., CCDC, Cambridge CB2 1EZ, UK) was applied for molecular docking calculations. Hydrogen atoms were added to the biomacromolecules following the ionization and tautomeric states at pH 7.4. Since there is no crystallographic structure for drugs associated with gD-nectin-1 and TFIIBc-VP16, the standard function *ChemPLP* was used for the molecular docking calculations by selecting a spherical cavity with a 10 Å radius around the active binding site. PyMOL Delano Scientific LLC software (Schrödinger, New York, NY, USA) was used to build the figures.

### 3.9. Statistical Analysis

All data presented in this paper represent the means ± standard deviation (SD) (SEM) of at least three independent experiments. The inhibition percentages are related to control assays with virus infection without treatment. All statistical analyses were performed using the software GraphPrism 8. Values of *p* ≤ 0.001 were considered statistically significant by Dunnett’s multiple comparisons test, one-way ANOVA (Figure 2 and Figure 3), or paired *t*-test (Figure 4).

## 4. Conclusions

The pyrazolopyridine derivatives evaluated here exhibited a new and different mechanism of inhibition when compared with standard anti-HSV-1 agents typically found in clinical practice, such as ACV, which inhibit viral DNA polymerase. Therefore, even mutations in DNA polymerase of HSV-1, which could produce resistance to the reference compounds, could not affect the action of the studied pyrazolopyridine derivatives. Moreover, we describe for the first time that the compound **AM-57** could inhibit different steps of viral replication, affecting critical viral proteins, such as ICP27 and gD, by targeting the viral complex between TFIIBc and VP16. We conclude that the constant design and study of novel antiviral compounds may provide a future basis for novel antiviral therapeutic molecules with unique structures and alternative mechanisms of action. Thus, we propose that the chemical structures of **ARA-04**, **ARA-05**, and **AM-57**, both derivatives of the 1*H*-pyrazolo[3,4-*b*]pyridine system, are promising for future bioisosterically broad-spectrum analysis for novel antivirals, with a good perspective for structure–activity relationship (SAR) studies.

## Figures and Tables

**Figure 1 ijms-23-08135-f001:**
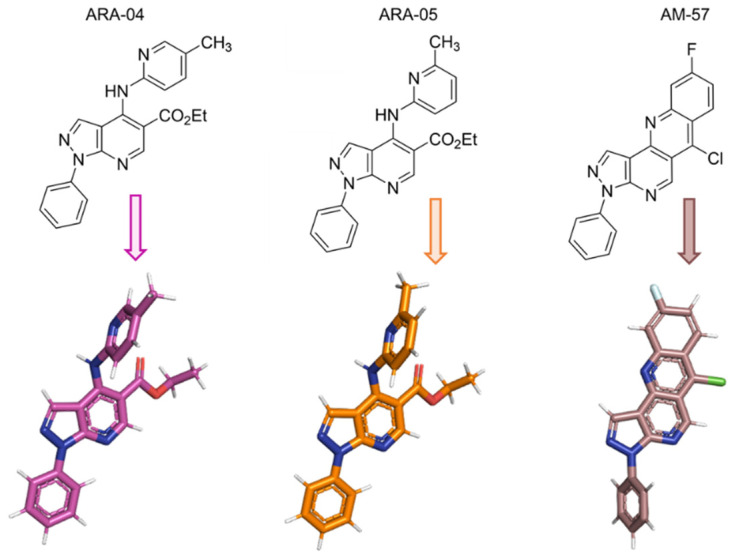
The 2D and 3D chemical structure of the studied compounds. Ethyl 4-((5-methylpyridin-2-yl)amino)-1-phenyl-1*H*-pyrazolo[3,4-*b*]pyridine-5-carboxylate (**ARA-04**), ethyl 4-((6-methylpyridin-2-yl)amino)-1-phenyl-1*H*-pyrazolo[3,4-*b*]pyridine-5-carboxylate (**ARA-05**), and 6-chloro-9-fluoro-3-phenyl-3*H*-benzo[*b*]pyrazolo[3,4-*h*][1,6]naphthyridine (**AM-57**).

**Figure 2 ijms-23-08135-f002:**
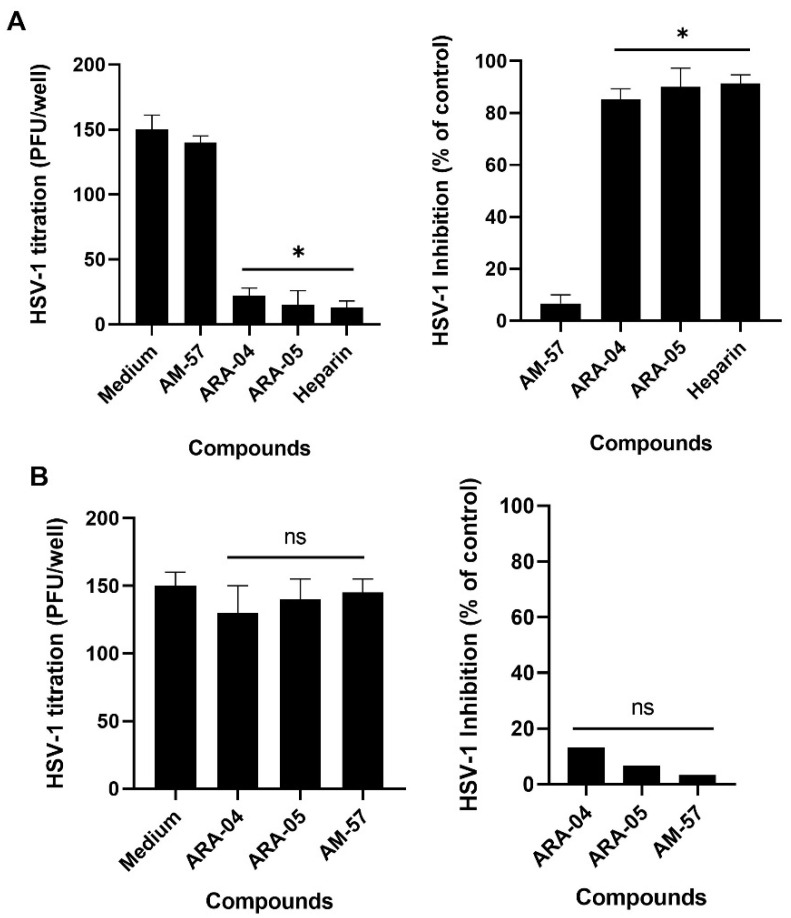
The compounds ARA-04 and ARA-05 inhibit the adsorption of HSV-1, but ARA-04, ARA-05, and AM-57 do not have an effect on HSV-1 penetration. (**A**) We infected Vero cells with 200 PFU/well of HSV-1 and treated them with the indicated compounds using their EC_50_ concentrations and heparin at 1 µg/mL for 1 h at 4 °C. After that, the inoculum was removed, cells were washed, covered with overlay medium, and the temperature was shifted to 37 °C for 72 h. Then, viral plaques were counted (*n* = 3). (**B**) Vero cells were infected with 200 PFU/well for 1 h at 4 °C. After that, the inoculum was removed, and cells were treated with the indicated compounds for an additional 1 h at 37 °C. Then, supernatants of the cell cultures were harvested, and cells were washed and covered with overlay medium. The viral plaques were counted 72 h post-infection (*n* = 3). * *p* ≤ 0.001, ns indicates “not significant”.

**Figure 3 ijms-23-08135-f003:**
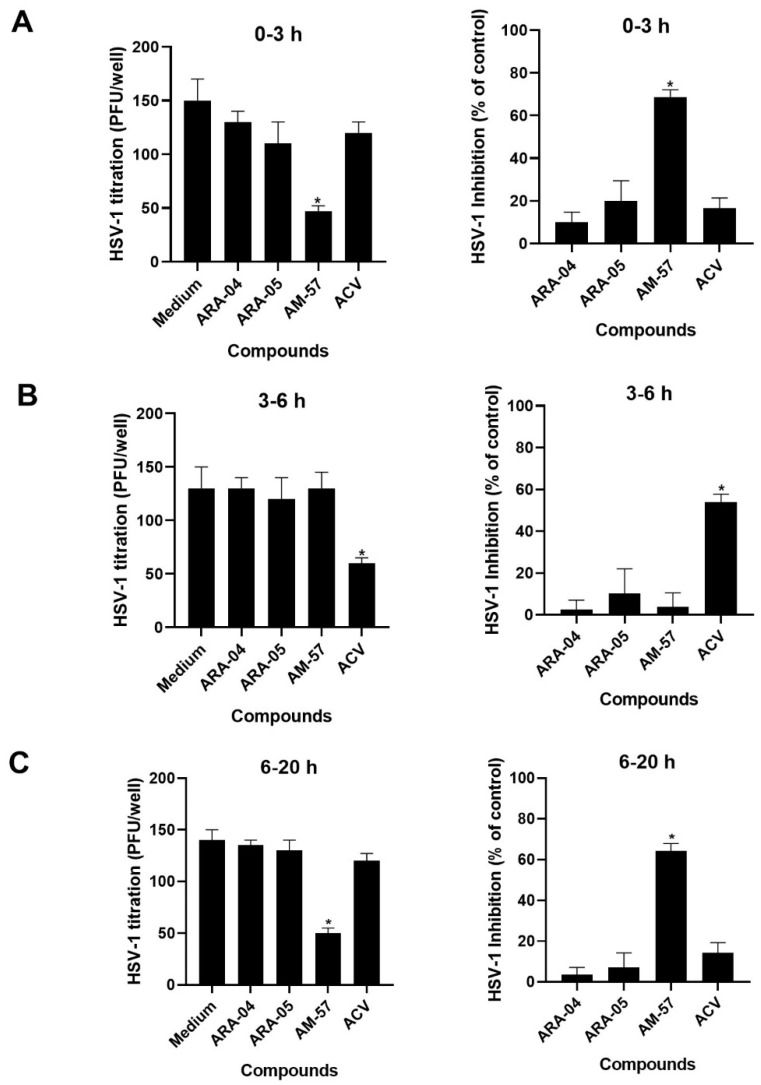
AM-57 inhibits HSV-1 during two moments of viral replication. Vero cells were infected with HSV-1 for 1 h at 37 °C (200 PFU/well) and treated with the indicated compounds at their EC_50_ concentrations. These treatments were performed at different time frames after infection: 0–3 (**A**), 3–6 (**B**)**,** and 6–20 (**C**). After each time, medium containing the drugs were removed and overlay medium was added after 72 h p.i., and viral plaques were counted (*n* = 3), * *p* ≤ 0.001.

**Figure 4 ijms-23-08135-f004:**
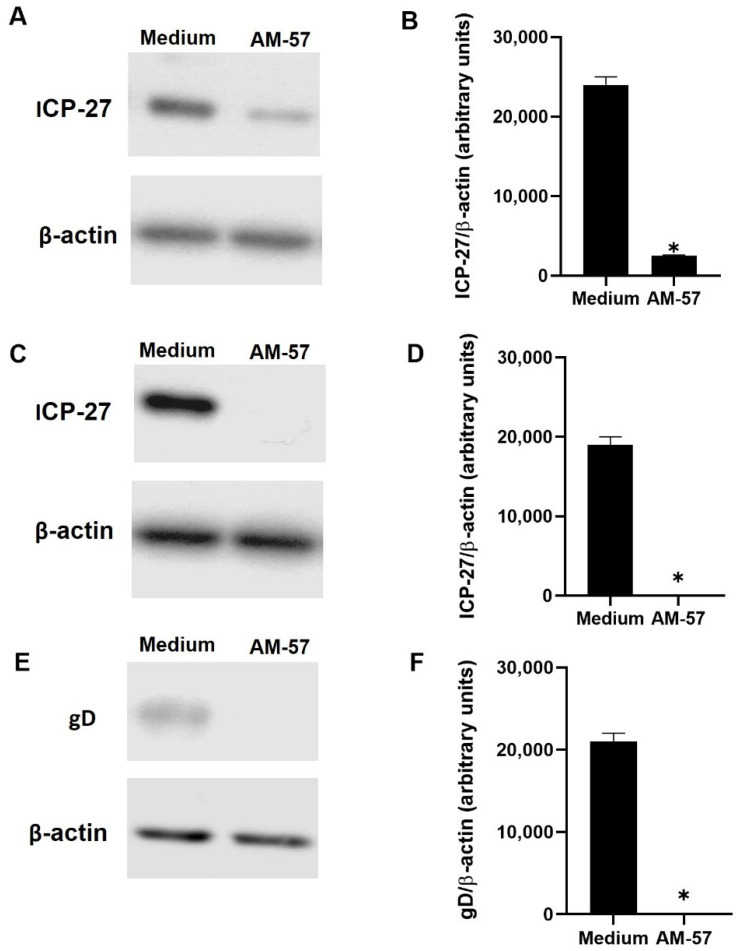
AM-57 interferes with the content of HSV-1 ICP27 and gD proteins. Vero cells were infected with HSV-1 at an MOI of 1 and treated with **AM-57** (0.7 µM) during the time frames from 0–3 h (**A**,**B**) to 6–20 h (**C**–**F**) after infection. After each period, cells were lysed and immunoblotted (**A**,**C**,**E**). The corresponding densitometry analysis is also displayed (**B**,**D**,**F**). The gels are representative of three independent experiments, * *p* < 0.001.

**Figure 5 ijms-23-08135-f005:**
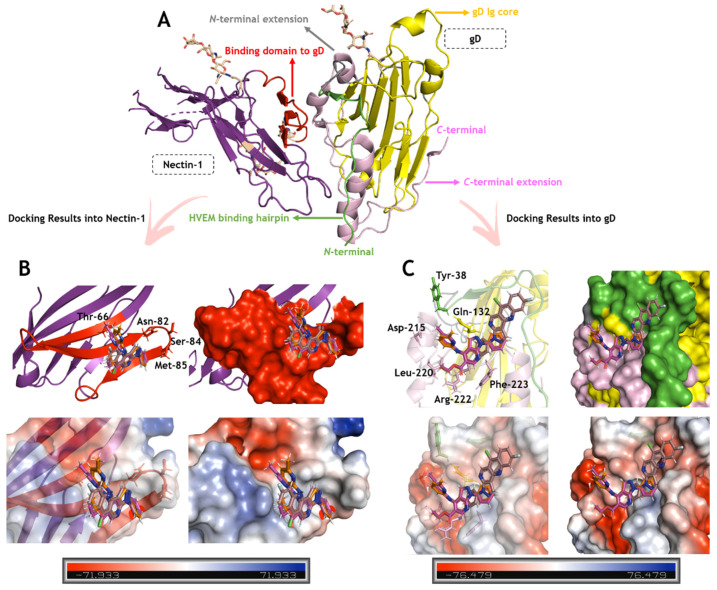
Binding pose of ARA-04, ARA-05, and AM-57 into the gD-nectin-1 structure. (**A**) Three-dimensional (3D) structure for gD-nectin-1 (PDB Code: 3SKU), highlighting the main structural motifs [53]. (**B**) Superposition of the in silico results for **ARA-04**, **ARA-05**, and **AM-57** into the nectin-1 motifs responsible for the interaction with gD (top: cartoon and surface representation; bottom: electrostatic potential map representation). (**C**) Superposition of the in silico results for **ARA-04**, **ARA-05**, and **AM-57** into the gD motifs responsible for the interaction with nectin-1 (top: cartoon and surface representation; bottom: electrostatic potential map representation). The key amino acids responsible for the interaction of gD-nectin-1 that interact with the compounds under study are in stick representation, while **ARA-04**, **ARA-05**, and **AM-57** are in stick representation in pink, orange, and brown, respectively. Element color for **ARA-04**, **ARA-05**, and **AM-57**: hydrogen, nitrogen, chloro, fluorine, and oxygen in white, dark blue, light green, cyan, and red, respectively.

**Figure 6 ijms-23-08135-f006:**
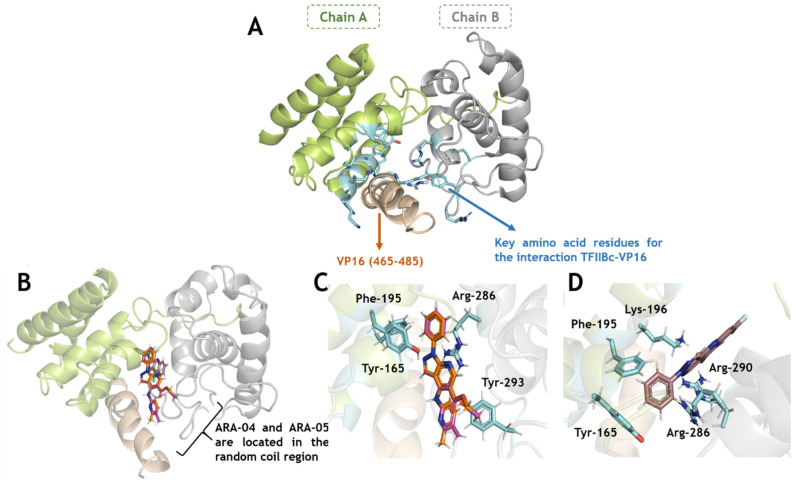
Binding pose of ARA-04, ARA-05, and AM-57 into the TFIIBc-VP16 structure. (**A**) Three-dimensional (3D) structure for TFIIBc-VP16 (PDB Code: 2PHG), highlighting the key amino acids of TFIIBc (in stick representation in cyan) that interact with the interactive motif of VP16 (in wheat color) [51]. (**B**) Superposition of the in silico results for **ARA-04**, **ARA-05**, and **AM-57** into the complex TFIIBc-VP16. (**C**) Best docking pose for **ARA-04** and **ARA-05** into the complex TFIIBc-VP16, highlighting the key amino acid residues of TFIIBc that interacted with VP16 and now interact with **ARA-04** and **ARA-05**. (**D**) In silico result for **AM-57** into the complex TFIIBc-VP16, highlighting the key amino acid residues of TFIIBc that interacted with VP16 and now interact with **AM-57**. The 3D structures of **ARA-04**, **ARA-05**, and **AM-57** are in stick representation in pink, orange, and brown, respectively. Element color for **ARA-04**, **ARA-05**, and **AM-57**: hydrogen, nitrogen, chloro, fluorine, and oxygen in white, dark blue, light green, cyan, and red, respectively.

**Table 1 ijms-23-08135-t001:** Cytotoxicity and anti-HSV-1 activity of the compounds **ARA-04**, **ARA-05**, and **AM-57**.

Compound	CC_50_ (μM)	EC_50_ (μM)	SI ^a^
**ARA-04**	1000 ± 100	1.00 ± 0.10	1000
**ARA-05**	1000 ± 80	1.00 ± 0.05	1000
**AM-57**	600 ± 50	0.70 ± 0.10	857
ACV	960 ± 156	1.10 ± 0.25	880

^a^ Calculated from the ratio of CC_50_ and EC_50._

**Table 2 ijms-23-08135-t002:** Docking score value (dimensionless) for the interaction of **ARA-04**, **ARA-05**, and **AM-57** with the biomacromolecules gD, nectin-1, TFIIBc-VP16, TFIIBc, and VP16.

Compound	gD	Nectin-1	TFIIBc-VP16	TFIIBc	VP16
**ARA-04**	55.8	46.8	8.39	42.7	39.9
**ARA-05**	55.9	41.1	5.84	43.2	42.5
**AM-57**	44.9	30.2	25.5	56.0	50.1

## Data Availability

All analyzed data are contained in the main text. Raw data are available from the authors upon request.

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
