# Peer review of "The Role of Pyrazolopyridine Derivatives on Different Steps of Herpes Simplex Virus Type-1 In Vitro Replicative Cycle"

_ijms, 2022, doi:10.3390/ijms23158135_

Round 1

Reviewer 1 Report

 Authors presented interesting results, that the studied derivatives inhibit HSV-1 replicative cycle with a novel mechanism of action and its scaffold can be used as a template for the synthesis of promising new molecules with antiviral effect.

I have one general remark. Authors studied 3 compounds with common core structure. Derivative AM-57 differs from two other structures. All conclusions are based on limited (via number of studied compounds) data. This novel mechanism should be confirmed by studing larger number of AM-57 derivatives. 

Please add SAR conclusions based on previous published results and related scientific papers. 

Author Response

We appreciate this referee’s evaluation of the manuscript. Additional series of compounds would be necessary to perform SAR analysis, they have been published elsewhere (doi: 10.1186/2191-2858-2-3). Even these derivatives may be not enough to determine which radicals/moieties are endowed with best antiviral activity against proposed mechanism of action from AM-57 (doi: 10.1021/ci0001177). We interpret this should be aim of a new investigation. Nevertheless, here, we present the discovery, based on cellular infection assays and computational methods, that AM-57 could have a novel mechanism of inhibition. Anyway, to avoid overselling of the findings in the manuscript in section 2.2, we revised the results and discussion to alleviate the conclusions.

Reviewer 2 Report

HSV establishes a latency state in an infected organism and may reactivate under the influence of various factors

The presented study is interesting. The aim of the authors' research was to find substances with antiviral activity against HSV. The emergence of HSV-resistant strains to the currently available drugs (acyclovir and its derivatives) forces researchers to look for new compounds with antiviral activity, especially with novel mechanisms  of action.  

The 1H-pyrazolo[3,4-b]pyridine system have anticancer, antimalarial, anti-inflammatory, analgesic, anti- bacterial, and anti-protozoan activities.  The antiviral activity has been described by authors in their previous study. The Authors named the studied compounds as ARA-04ARA-05, and AM-57 

Cytotoxic and antiviral effect was studied in Vero cells. The statistical analyses were performed 407 using the software Graph Prism 8.  

Their results indicated that ARA-04 and ARA-05 affect viral adsorption, while AM-57 interferes the virus replication during its α and γ-phases and decreased ICP27 content during initial and late events of HSV-1 replication.  

ICP27 is a regulatory protein and play a role in the expression of HSV-1 γ genes, including the viral gD.  Inhibition of the gD expression can play a key role in block of viral fusion with the host cell. . 

The Authors suggest that ARA-04ARA-05, and AM-57may be promising for future analyzes of new antiviral drugs.

Author Response

We appreciate the referee's evaluation of our manuscript. As the present report had no queries, we assumed that this referee classified the article as ready to be published, which means that we did some few changes just to satisfy the other referee´s demands.

Reviewer 3 Report

Miranda et al. in the manuscript „The role of pyrazolopyridine derivates…“describe antiviral properties of pyrazolopyridine derivates against HSV-1, dsDNA virus. The group has previously shown the activity of the same compound on vaccinia virus. Interestingly, the tested compounds show effectiveness to inhibit virus replication, and cytotoxicity similar to, similar to ACV. Authors, based on several assays, and in silico cellulations, suggest targets of three different compounds. The manuscript is written in good English, relatively easy to read and pertinent in the context of novel antiviral-drug research.

The main straight of the manuscript relays on novel compounds which show promising activity, however drown conclusions are premature; i.e. authors do not support these with sufficient evidences. Particularly, the molecular mechanisms are lacking more detailed analysis: e.g. penetration/adsorption assays would benefit with RTqPCR (HSV-1 DNA). Authors are not considering different mechanisms of action in specific assays; for example: the inhibition of attachment/penetration might result from post-entry events. Overall, authors show that the tested compound might inhibit virus replication by many/different mechanisms, however, the statement are strongly overstated.  Authors should reconceptualize this important work.

Author Response

Please see our changes throughout the manuscript. The reviewer is right, and we alleviate several conclusions. We also understand that qPCR for HSV-1 DNA could have been used as the read out for the adsorption and penetration assays, but only if the same compound would be able to inhibit entry and the subsequent steps of virus replication. Whereas, AM-57 inhibits the alpha/gamma phases of HSV-1 replication, it did not inhibit viral adsorption or penetration. Conversely, ARA-related compounds inhibit entry, but did not affect the next phases of replication. Thus, the adsorption and entry assays using temperature shift and the PFU as the experiment readout, as previously investigated by our group and others (10.1055/s-0029-1186144; 10.1016/j.antiviral.2007.08.011; 10.1007/s00705-007-0960-y; 10.1055/s-2007-967109; 10.1016/s0166-3542(02)00054-2; 10.3389/fmicb.2021.736780), will not be influenced by the effect of the ARA-related compounds on the next steps of virus life cycle. Moreover, our focus on AM-57 may allow us to leave the subsequent results from Figure 3 henceforward as they are.

Round 2

Reviewer 3 Report

I thank the authors for their response and their efforts to justify publication in the IJMS (IF > 6). The main weaknesses of the manuscript remain. There is no evidence for the suggested mechanisms but in silico predictions for selected targets. The observed downregulation of ICP27 is likely to be seen with almost any drug that affects viral replication before DNA replication, and therefore the result can only be circumstantial because of the large number of possible mechanisms. The same is true for gD. It is reasonable to assume that the levels of all immediate early viral proteins follow the ICP27 pattern, and therefore blithely selected targets based on the simple statement "we decided to investigate whether the drug AM-57 could affect the levels of some of the major HSV-1 proteins, such as ICP27 and gD" cannot be justified. If some sort of screening of other potential targets or additional evidence of specific binding were presented, a different conclusion might be reached.

Author Response

We agree with the Reviewer that AM-57 is not interacting and inhibiting directly ICP27 and/or gD. These proteins were used as markers of α and γ steps of viral replication (Figure 4), to further characterize the temporal inhibition described in Figure 3. Because we agree, in silico analysis with AM-57 were performed at a more upstream event of virus life cycle, the TFIIBc-VP16 structure. Please see our changes in blue throughout this manuscript, where our writing was adjusted to be convergente with these ideas and also were the limitations are stated.  
